# Dual Variational Generation for Low Shot Heterogeneous Face Recognition

**Chaoyou Fu**[1,2]*, **Xiang Wu**[1]*, **Yibo Hu**[1], **Huaibo Huang**[1], **Ran He**[1,2,3]†
[1]NLPR & CRIPAC, CASIA
[2]University of Chinese Academy of Sciences
[3]Center for Excellence in Brain Science and Intelligence Technology, CAS
{chaoyou.fu, rhe}@nlpr.ia.ac.cn, alfredxiangwu@gmail.com
{yibo.hu, huaibo.huang}@cripac.ia.ac.cn

## Abstract

Heterogeneous Face Recognition (HFR) is a challenging issue because of the large domain discrepancy and a lack of heterogeneous data. This paper considers HFR as a dual generation problem, and proposes a novel Dual Variational Generation (DVG) framework. It generates large-scale new paired heterogeneous images with the same identity from noise, for the sake of reducing the domain gap of HFR. Specifically, we first introduce a dual variational autoencoder to represent a joint distribution of paired heterogeneous images. Then, in order to ensure the identity consistency of the generated paired heterogeneous images, we impose a distribution alignment in the latent space and a pairwise identity preserving in the image space. Moreover, the HFR network reduces the domain discrepancy by constraining the pairwise feature distances between the generated paired heterogeneous images. Extensive experiments on four HFR databases show that our method can significantly improve state-of-the-art results.

## 1 Introduction

With the development of deep learning, face recognition has made significant progress [34, 2] in recent years. However, in many real-world applications, such as video surveillance, facial authentication on mobile devices and computer forensics, it is still a great challenge to match heterogeneous face images in different modalities, including sketch images [37], near infrared images [24] and polarimetric thermal images [36]. Heterogeneous face recognition (HFR) has attracted much attention in the face recognition community. Due to the large domain gap, one challenge is that the face recognition model trained on VIS data often degrades significantly for HFR. Therefore, lots of cross domain feature matching methods [10] are introduced to reduce the large domain gap between heterogeneous face images. However, since it is expensive and time-consuming to collect a large number of heterogeneous face images, there is no public large-scale heterogeneous face database. With the limited training data, CNNs trained for HFR often tend to be overfitting.

Recently, the great progress of high-quality face synthesis [38, 5, 33, 39] has made "recognition via generation" possible. TP-GAN [16] and CAPG-GAN [13] introduce face synthesis to improve the quantitative performance of large pose face recognition. For HFR, [32] proposes a two-path model to synthesize VIS images from NIR images. [36] utilizes a GAN based multi-stream feature fusion technique to generate VIS images from polarimetric thermal faces. However, all these methods are based on conditional image-to-image translation framework, leading to two potential challenges: 1)

---

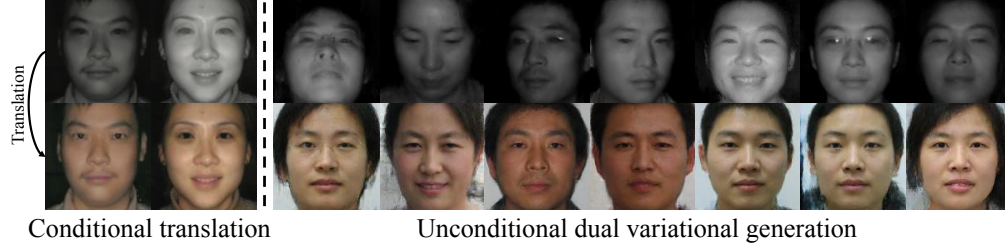

Conditional translation       Unconditional dual variational generation

Figure 1: The diversity comparisons between the conditional image-to-image translation [32] (left part, the above is the input NIR image and the below is the corresponding translated VIS image) and our unconditional DVG (right part, all paired heterogeneous images are generated via noise). For the conditional image-to-image translation methods, given one NIR image, a generator only synthesizes one new VIS image with same attributes (e.g., the pose and the expression) except for the spectral information. Differently, DVG generates massive new paired images with rich intra-class diversity from noise.

Diversity: Given one image, a generator only synthesizes one new image of the target domain [32]. It means such conditional image-to-image translation methods can only generate limited number of images. In addition, as shown in the left part of Fig. 1, two images before and after translation have same attributes (e.g., the pose and the expression) except for the spectral information, which means it is difficult for such conditional image-to-image translation methods to promote intra-class diversity. In particular, these problems will be very prominent in the low-shot heterogeneous face recognition, i.e., learning from few heterogeneous data. 2) Consistency: When generating large-scale samples, it is challenging to guarantee that the synthesized face images belong to the same identity of the input images. Although identity preserving loss [13] constrains the distances between features of the input and synthesized images, it does not constraint the intra-class and inter-class distances of the embedding space.

To tackle the above challenges, we propose a novel unconditional Dual Variational Generation (DVG) framework (shown in Fig. 3) that generates large-scale paired heterogeneous images with the same identity from noise. Unconditional generative models can generate new images (generate single image per time) from noise [21], but since these images do not have identity labels, it is difficult to use these images for recognition networks. DVG makes use of the property of generating new images of the unconditional generative model [21], and adopts a dual generation manner to get paired heterogeneous images with the same identity every time. This enables DVG to generate large-scale images, and make the generated images can be used to optimize recognition networks. Meanwhile, DVG also absorbs the various intra-class changes of the training database, leading to the generated paired images have abundant intra-class diversity. For instance, as presented in the right part of Fig. 1, the first four paired images have different poses, and the fifth paired images have different expressions. Furthermore, DVG only pays attention to the identity consistency of the paired heterogeneous images rather than the identity whom the paired heterogeneous images belong to, which avoids the consistency problem of previous methods. Specifically, we introduce a dual variational autoencoder to learn a joint distribution of paired heterogeneous images. In order to constrain the generated paired images to belong to the same identity, we impose both a distribution alignment in the latent space and a pairwise identity preserving in the image space. New paired images are generated by sampling and copying a noise from a standard Gaussian distribution, as displayed in the left part of Fig. 3. These generated paired images are used to optimize the HFR network by a pairwise distance constraint, aiming at reducing the domain discrepancy.

In summary, the main contributions are as follows:

- We provide a new insight into the problems of HFR. That is, we consider HFR as a dual generation problem, and propose a novel dual variational generation framework. This framework generates new paired heterogeneous images with abundant intra-class diversity to reduce the domain gap of HFR.

- In order to guarantee that the generated paired images belong to the same identity, we constrain the consistency of paired images in both latent space and image space. These allow new images sampled from the noise can be used for recognition networks.

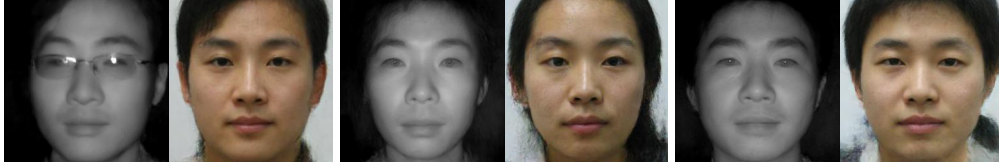

Figure 2: The dual generation results via noise ($256 \times 256$ resolution). For each pair, the left is NIR and the right is the paired VIS image.

- We can sample large-scale diverse paired heterogeneous images from noise. By constraining the pairwise feature distances of the generated paired images in the HFR network, the domain discrepancy is effectively reduced.
- Experiments on the CASIA NIR-VIS 2.0, the Oulu-CASIA NIR-VIS, the BUAA-VisNir and the IIIT-D Viewed Sketch databases demonstrate that our method can generate photo-realistic images, and significantly improve the performance of recognition.

## 2 Background and Related Work

### 2.1 Heterogeneous Face Recognition

Lots of researchers pay their attention to Heterogeneous Face Recognition (HFR). For the feature-level learning, [22] employs HOG features with sparse representation for HFR. [7] utilizes LBP histogram with Linear Discriminant Analysis to obtain domain-invariant features. [10] proposes Invariant Deep Representation (IDR) to disentangle representations into two orthogonal subspaces for NIR-VIS HFR. Further, [11] extends IDR by introducing Wasserstein distance to obtain domain invariant features for HFR. For the image-level learning, the common idea is to transform heterogeneous face images from one modality into another one via image synthesis. [19] utilizes joint dictionary learning to reconstruct face images for boosting the performance of face matching. [23] proposes a cross-spectral hallucination and low-rank embedding to synthesize a heterogeneous image in a patch way.

### 2.2 Generative Models

Variational autoencoders (VAEs) [21] and generative adversarial networks (GANs) [6] are the most prominent generative models. VAEs consist of an encoder network $q_\phi(z|x)$ and a decoder network $p_\theta(x|z)$. $q_\phi(z|x)$ maps input images $x$ to the latent variables $z$ that match to a prior $p(z)$, and $p_\theta(x|z)$ samples images $x$ from the latent variables $z$. The evidence lower bound objective (ELBO) of VAEs:

$$\log p_\theta(x) \geq \mathbb{E}_{q_\phi(z|x)} \log p_\theta(x|z) - D_{\mathrm{KL}}(q_\phi(z|x)||p(z)). \tag{1}$$

The two parts in ELBO are a reconstruction error and a Kullback-Leibler divergence, respectively.

Differently, GANs adopt a generator $G$ and a discriminator $D$ to play a min-max game. $G$ generates images from a prior $p(z)$ to confuse $D$, and $D$ is trained to distinguish between generated data and real data. This adversarial rule takes the form:

$$\min_G \max_D \mathbb{E}_{x \sim p_{data}(x)} \left[\log D(x)\right] + \mathbb{E}_{z \sim p_z(z)} \left[\log(1 - D(G(z)))\right]. \tag{2}$$

They have achieved remarkable success in various applications, such as unconditional image generation that generates images from noise [20, 15], and conditional image generation that synthesizes images according to the given condition [32, 16]. According to [15], VAEs have nice manifold representations, while GANs are better at generating sharper images.

Another work to address the similar problem of our method is CoGAN [25], which uses a weight-sharing manner to generate paired images in two different modalities. However, CoGAN neither explicitly constrains the identity consistency of paired images in the latent space nor in the image space. It is challenging for the weight-sharing manner of CoGAN to generate paired images with the same identity, as shown in Fig. 4.

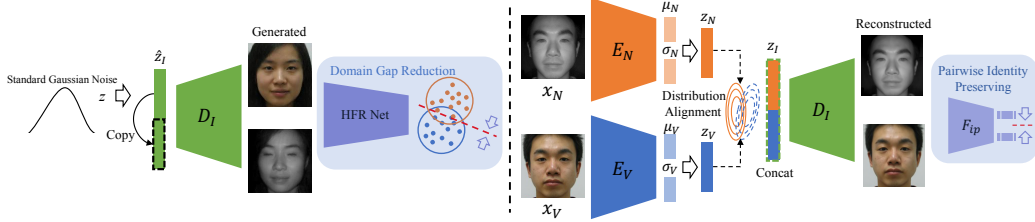

Figure 3: The purpose (left part) and training model (right part) of our unconditional DVG framework. DVG generates large-scale new paired heterogeneous images with the same identity from standard Gaussian noise, aiming at reducing the domain discrepancy for HFR. In order to achieve this purpose, we elaborately design a dual variational autoencoder. Given a pair of heterogeneous images from the same identity, the dual variational autoencoder learns a joint distribution in the latent space. In order to guarantee the identity consistency of the generated paired images, we impose a distribution alignment in the latent space and a pairwise identity preserving in the image space.

## 3 Proposed Method

In this section, we will introduce our method in detail, including the dual variational generation and heterogeneous face recognition. Note that we specifically discuss the NIR-VIS images for better presentation. Other heterogeneous images are also applicable.

### 3.1 Dual Variational Generation

As shown in the right part of Fig. 3, DVG consists of a feature extractor $F_{ip}$, and a dual variational autoencoder: two encoder networks and a decoder network, all of which play the same roles of VAEs [21]. Specifically, $F_{ip}$ extracts the semantic information of the generated images to preserve the identity information. The encoder network $E_N$ maps NIR images $x_N$ to a latent space $z_N = q_{\phi_N}(z_N|x_N)$ by a reparameterization trick: $z_N = u_N + \sigma_N \odot \epsilon$, where $u_N$ and $\sigma_N$ denote mean and standard deviation of NIR images, respectively. In addition, $\epsilon$ is sampled from a multi-variate standard Gaussian and $\odot$ denotes the Hadamard product. The encoder network $E_V$ has the same manner as $E_N$: $z_V = q_{\phi_V}(z_V|x_V)$, which is for VIS images $x_V$. After obtaining the two independent distributions, we concatenate $z_N$ and $z_V$ to get the joint distribution $z_I$.

**Distribution Learning** We utilize VAEs to learn the joint distribution of the paired NIR-VIS images. Given a pair of NIR-VIS images $\{x_N, x_V\}$, we constrain the posterior distribution $q_{\phi_N}(z_N|x_N)$ and $q_{\phi_V}(z_V|x_V)$ by the Kullback-Leibler divergence:

$$\mathcal{L}_{\text{kl}} = D_{\text{KL}}(q_{\phi_N}(z_N|x_N)||p(z_N)) + D_{\text{KL}}(q_{\phi_V}(z_V|x_V)||p(z_V)), \tag{3}$$

where the prior distributions $p(z_N)$ and $p(z_V)$ are both the multi-variate standard Gaussian distributions. Like the original VAEs, we require the decoder network $p_\theta(x_N, x_V|z_I)$ to be able to reconstruct the input images $x_N$ and $x_V$ from the learned distribution:

$$\mathcal{L}_{\text{rec}} = -\mathbb{E}_{q_{\phi_N}(z_N|x_N) \cup q_{\phi_V}(z_V|x_V)} \log p_\theta(x_N, x_V|z_I). \tag{4}$$

**Distribution Alignment** We expect a pair of NIR-VIS images $\{x_N, x_V\}$ to be projected into a common latent space by the encoders $E_N$ and $E_V$, i.e., the NIR distribution $p(z_N^{(i)})$ is the same as the VIS distribution $p(z_V^{(i)})$, where $i$ denotes the identity information. That means we maintain the identity consistency of the generated paired images in the latent space. Explicitly, we align the NIR and VIS distributions by minimizing the Wasserstein distance between the two distributions. Given two Gaussian distributions $p(z_N^{(i)}) = N(u_N^{(i)}, \sigma_N^{(i)^2})$ and $p(z_V^{(i)}) = N(u_V^{(i)}, \sigma_V^{(i)^2})$, the 2-Wasserstein distance between $p(z_N^{(i)})$ and $p(z_V^{(i)})$ is simplified [10] as:

$$\mathcal{L}_{\text{dist}} = \frac{1}{2} \left[ ||u_N^{(i)} - u_V^{(i)}||_2^2 + ||\sigma_N^{(i)} - \sigma_V^{(i)}||_2^2 \right]. \tag{5}$$

**Pairwise Identity Preserving**   In previous image-to-image translation works [16, 13], identity preserving is usually introduced to maintain identity information. The traditional approach uses a pre-trained feature extractor to enforce the features of the generated images to be close to the target ones. However, since the lack of intra-class and inter-class constraints, it is challenge to guarantee the synthesized images to belong to the specific categories of the target images. Considering that DVG generates a pair of heterogeneous images per time, we only need to consider the identity consistency of the paired images.

Specifically, we adopt Light CNN [34] as the feature extractor $F_{ip}$ to constrain the feature distance between the reconstructed paired images:

$$\mathcal{L}_{\text{ip-pair}} = ||F_{ip}(\hat{x}_N) - F_{ip}(\hat{x}_V)||_2^2, \tag{6}$$

where $F_{ip}(\cdot)$ means the normalized output of the last fully connected layer of $F_{ip}$. In addition, we also use $F_{ip}$ to make the features of the reconstructed images and the original input images close enough as previous works [16, 13]:

$$\mathcal{L}_{\text{ip-rec}} = ||F_{ip}(\hat{x}_N) - F_{ip}(x_N)||_2^2 + ||F_{ip}(\hat{x}_V) - F_{ip}(x_V)||_2^2, \tag{7}$$

where $\hat{x}_N$ and $\hat{x}_V$ denote the reconstructions of the input paired images $x_N$ and $x_V$, respectively.

**Diversity Constraint**   In order to further increase the diversity of the generated images, we also introduce a diversity loss [27]. In the sampling stage, when two sampled noise $z_{I_1}$ are $z_{I_2}$ are close, the generated images $x_{I_1}$ and $x_{I_2}$ are going to be similar. We maximize the following loss to encourage the decoder $D_I$ to generate more diverse images:

$$\mathcal{L}_{\text{div}} = \max_{D_I} \frac{|F_{ip}(x_{I_1}) - F_{ip}(x_{I_2})|}{|z_{I_1} - z_{I_2}|}. \tag{8}$$

**Overall Loss**   Moreover, in order to increase the sharpness of our generated images, we also adopt an adversarial loss $\mathcal{L}_{\text{adv}}$ as [31]. Hence, the overall loss to optimize the dual variational autoencoder can be formulated as

$$\mathcal{L}_{\text{gen}} = \mathcal{L}_{\text{rec}} + \mathcal{L}_{\text{kl}} + \mathcal{L}_{\text{adv}} + \lambda_1 \mathcal{L}_{\text{dist}} + \lambda_2 \mathcal{L}_{\text{ip-pair}} + \lambda_3 \mathcal{L}_{\text{ip-rec}} + \lambda_4 \mathcal{L}_{\text{div}}, \tag{9}$$

where $\lambda_1$, $\lambda_2$, $\lambda_3$ and $\lambda_4$ are the trade-off parameters.

### 3.2   Heterogeneous Face Recognition

For the heterogeneous face recognition, our training data contains the original limited labeled data $x_i(i \in \{N, V\})$ and the large-scale generated unlabeled paired NIR-VIS data $\tilde{x}_i(i \in \{N, V\})$. Here, we define a heterogeneous face recognition network $F$ to extract features $f_i = F(x_i; \Theta)$, where $i \in \{N, V\}$ and $\Theta$ is the parameters of $F$. For the original labeled NIR and VIS images, we utilize a softmax loss:

$$\mathcal{L}_{\text{cls}} = \sum_{i \in \{N,V\}} \text{softmax}(F(x_i; \Theta), y), \tag{10}$$

where $y$ is the label of identity.

For the generated paired heterogeneous images, since they are generated from noise, there are no specific classes for the paired images. But as mentioned in section 3.1, DVG ensures that the generated paired images belong to the same identity. Therefore, a pairwise distance loss between the paired heterogeneous samples is formulated as follows:

$$\mathcal{L}_{\text{pair}} = ||F(\tilde{x}_N; \Theta) - F(\tilde{x}_V; \Theta)||_2^2, \tag{11}$$

In this way, we can efficiently minimize the domain discrepancy by generating large-scale unlabeled paired heterogeneous images. As stated above, the final loss to optimize for the heterogeneous face recognition network can be written as

$$\mathcal{L}_{\text{hfr}} = \mathcal{L}_{\text{cls}} + \alpha_1 \mathcal{L}_{\text{pair}}, \tag{12}$$

where $\alpha_1$ is the trade-off parameter.

| Method | MD | FID | Rank-1 |
|--------|------|------|------|
| CoGAN | 0.61 | 10.6 | 95.2 |
| VAE | 0.54 | 8.2 | 94.6 |
| DVG | **0.24** | **7.1** | **99.2** |

| Method | Rank-1 |
|--------|--------|
| w/o $\mathcal{L}_{\text{dist}}$ | 94.3 |
| w/o $\mathcal{L}_{\text{ip-pair}}$ | 96.1 |
| w/o $\mathcal{L}_{\text{div}}$ | 98.5 |
| DVG | **99.2** |

(a)                                    (b)

Table 1: Experimental analyses on the CASIA NIR-VIS 2.0 database. The backbone is LightCNN-9. (a) The quantitative comparisons of different methods. MD (lower is better) means the mean feature distance between the generated paired NIR and VIS images. FID (lower is better) is measured based on the features of LightCNN-9, instead of the traditional Inception model. (b) The ablation study.

## 4   Experiments

### 4.1   Databases and Protocols

Three NIR-VIS heterogeneous face databases and one Sketch-Photo heterogeneous face database are used to evaluate our proposed method. For the NIR-VIS face recognition, following [35], we report Rank-1 accuracy and verification rate (VR)@false accept rate (FAR) for the CASIA NIR-VIS 2.0 [24], the Oulu-CASIA NIR-VIS [18] and the BUAA-VisNir Face [14] databases. Note that, for the Oulu-CASIA NIR-VIS database, there are only 20 subjects are selected as the training set. In addition, the IIIT-D Viewed Sketch database [1] is employed for the Sketch-Photo face recognition. Due to the few number of images in the IIIT-D Viewed Sketch database, following the protocols of [3], we use the CUHK Face Sketch FERET (CUFSF) [37] as the training set and report the Rank-1 accuracy and VR@FAR=1% for comparisons.

### 4.2   Experimental Details

For the dual variational generation, the architectures of the encoder and decoder networks are the same as [15], and the architecture of our discriminator is the same as [31]. These networks are trained using Adam optimizer with a fixed rate of $0.0002$. Other parameters $\lambda_1$, $\lambda_2$, $\lambda_3$ and $\lambda_4$ in Eq. (9) are set to 50, 5, 1000 and 0.2, respectively. For the heterogeneous face recognition, we utilize both LightCNN-9 and LightCNN-29 [34] as the backbones. The models are pre-trained on the MS-Celeb-1M database [9] and fine-tuned on the HFR training sets. All the face images are aligned to $144 \times 144$ and randomly cropped to $128 \times 128$ as the input for training. Stochastic gradient descent (SGD) is used as the optimizer, where the momentum is set to 0.9 and weight decay is set to $5e$-4. The learning rate is set to $1e$-3 initially and reduced to $5e$-4 gradually. The batch size is set to 64 and the dropout ratio is 0.5. The trade-off parameters $\alpha_1$ in Eq. (12) is set to 0.001 during training.

### 4.3   Experimental Analyses

In this section, we analyze three metrics, including identity consistency, distribution consistency and visual quality, to demonstrate the effectiveness of DVG. The compared methods include CoGAN [25] and VAE [21]. For VAE model, the input is the concatenated NIR-VIS images.

**Identity Consistency.**   In order to analyze the identity consistency, we measure the feature distance between the generated paired images on the CASIA NIR-VIS 2.0 database. Specifically, we first use a pre-trained Light CNN-9 [34] to extract features and then measure the mean distance (MD) of the paired images. The results are reported in Table 1a. MD is computed from 50K generated image pairs and the MD value of the original database is $0.26$. We can clearly see that the MD value of DVG is even smaller than the original database, which means that our method can effectively guarantee the identity consistency of the generated paired images. The recognition performance of different methods is also reported in Table 1a. We can see that DVG correspondingly achieves the best results.

**Distribution Consistency.**   On the CASIA NIR-VIS 2.0 database, we take Fréchet Inception Distance (FID) [12] to measure the Fréchet distance of two distributions in the feature space, reflecting the distribution consistency. We first measure the FID between the generated VIS images and the real VIS images, and the FID between the generated NIR images and the real NIR images, respectively.

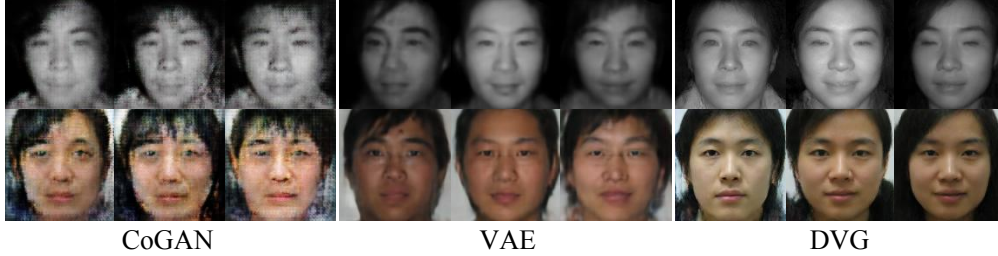

CoGAN                    VAE                     DVG

Figure 4: Visual comparisons of dual image generation results on the CASIA NIR-VIS 2.0 database. The generated paired images of DVG are more similar than those of CoGAN and VAE.

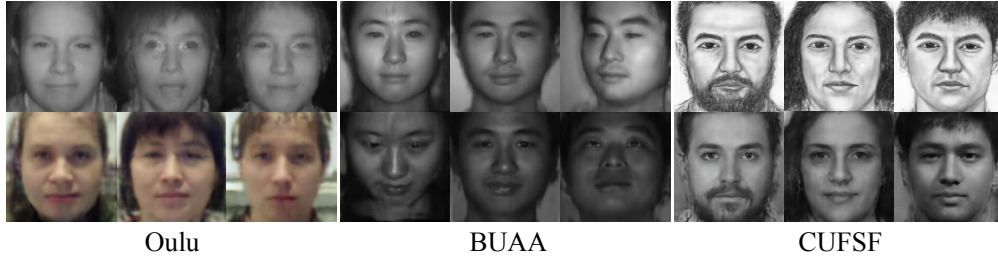

Oulu                    BUAA                    CUFSF

Figure 5: The dual generation results on the Oulu-CASIA NIR-VIS, the BUAA-VisNir and the CUHK Face Sketch FERET (CUFSF) databases.

Then we calculate the mean FID as the final results, which are reported in Table 1a. Considering that the face recognition network can better extract features of face images, we use a LightCNN-9 to extract features for calculating FID instead of the traditional Inception model. Similarly, FID results are computed from 50K generated image pairs. As shown in Table 1a, DVG achieves best results, demonstrating that DVG has really learned the distributions of two modalities.

**Visual Quality.**    In Fig. 4, we compare the dual generation results ($128 \times 128$ resolution) of different methods on the CASIA NIR-VIS 2.0 database. Our visual results are obviously better than CoGAN and VAE. Moreover, we can observe that the generated paired images of VAE and CoGAN are not similar, which leads to worse Rank-1 accuracy during optimizing HFR network (see Table 1a). More dual generation results of DVG are shown in Fig. 2 ($256 \times 256$ resolution) and Fig. 5.

**Ablation Study.**    Table 1b presents the comparison results of our DVG and its three variants on the CASIA NIR-VIS 2.0 database. We observe that the recognition performance will decrease if one component is not adopted. Particularly, the accuracy drops significantly when the distribution alignment loss $\mathcal{L}_{\text{dist}}$ or the pairwise identity preserving loss $\mathcal{L}_{\text{ip-pair}}$ are not used. These results suggest that every component is crucial in our model.

Moreover, we analyze how the number of generated samples influence the HFR network on the Oulu-CASIA NIR-VIS database that only contains 20 identities with about 1,000 images for training. We generate 1K, 5K, 10K and 50K pairs of heterogeneous images via DVG, and we obtain 68.7%, 85.9%, 89.5% and 89.4% on VR@FAR=0.1% by LightCNN-9, respectively. The results have been significantly improved with the increasing number of the generated pairs, suggesting that DVG can boost the performance of the low-shot heterogeneous face recognition.

## 4.4   Comparisons with State-of-the-art Methods

The recognition performance of our proposed DVG is demonstrated in this section on four heterogeneous face recognition databases. The performance of state-of-the-art methods, such as IDNet [29], HFR-CNN [30], Hallucination [23], DLFace [28], TRIVET [26], IDR [10], W-CNN [11], RCN [4], MC-CNN [3] and DVR [35] is compared in Table 2. In addition, LightCNN-9 and LightCNN-29 are our baseline methods.

| Method | CASIA NIR-VIS 2.0 | | Oulu-CASIA NIR-VIS | | | BUAA-VisNir | | | IIIT-D Viewed Sketch | |
|---|---|---|---|---|---|---|---|---|---|---|
| | Rank-1 | FAR=0.1% | Rank-1 | FAR=1% | FAR=0.1% | Rank-1 | FAR=1% | FAR=0.1% | Rank-1 | FAR=1% |
| IDNet [29] | $87.1 \pm 0.9$ | 74.5 | - | - | - | - | - | - | - | - |
| HFR-CNN [30] | $85.9 \pm 0.9$ | 78.0 | - | - | - | - | - | - | - | - |
| Hallucination [23] | $89.6 \pm 0.9$ | - | - | - | - | - | - | - | - | - |
| DLFace [28] | 98.68 | | - | - | - | - | - | - | - | - |
| TRIVET [26] | $95.7 \pm 0.5$ | $91.0 \pm 1.3$ | 92.2 | 67.9 | 33.6 | 93.9 | 93.0 | 80.9 | - | - |
| IDR [10] | $97.3 \pm 0.4$ | $95.7 \pm 0.7$ | 94.3 | 73.4 | 46.2 | 94.3 | 93.4 | 84.7 | - | - |
| W-CNN [11] | $98.7 \pm 0.3$ | $98.4 \pm 0.4$ | 98.0 | 81.5 | 54.6 | 97.4 | 96.0 | 91.9 | - | - |
| DVR [35] | $99.7 \pm 0.1$ | $99.6 \pm 0.3$ | 100.0 | 97.2 | 84.9 | 99.2 | **98.5** | 96.9 | - | - |
| RCN [4] | $99.3 \pm 0.2$ | $98.7 \pm 0.2$ | - | - | - | - | - | - | 90.34 | - |
| MC-CNN [3] | $99.4 \pm 0.1$ | $99.3 \pm 0.1$ | - | - | - | - | - | - | 87.40 | - |
| LightCNN-9 | $97.1 \pm 0.7$ | $93.7 \pm 0.8$ | 93.8 | 80.4 | 43.8 | 94.8 | 94.3 | 83.5 | 84.07 | 75.30 |
| LightCNN-9 + DVG | $99.2 \pm 0.3$ | $98.8 \pm 0.3$ | 100.0 | 97.6 | 89.5 | 98.0 | 97.1 | 93.1 | 86.65 | 92.24 |
| LightCNN-29 | $98.1 \pm 0.4$ | $97.4 \pm 0.5$ | 99.0 | 93.1 | 68.3 | 96.8 | 97.0 | 89.4 | 83.24 | 81.04 |
| LightCNN-29 + DVG | $\mathbf{99.8 \pm 0.1}$ | $\mathbf{99.8 \pm 0.1}$ | **100.0** | **98.5** | **92.9** | **99.3** | **98.5** | **97.3** | **96.99** | **97.86** |

Table 2: Comparisons with other state-of-the-art deep HFR methods on the CASIA NIR-VIS 2.0, the Oulu-CASIA NIR-VIS, the BUAA-VisNir and the IIIT-D Viewed Sketch databases.

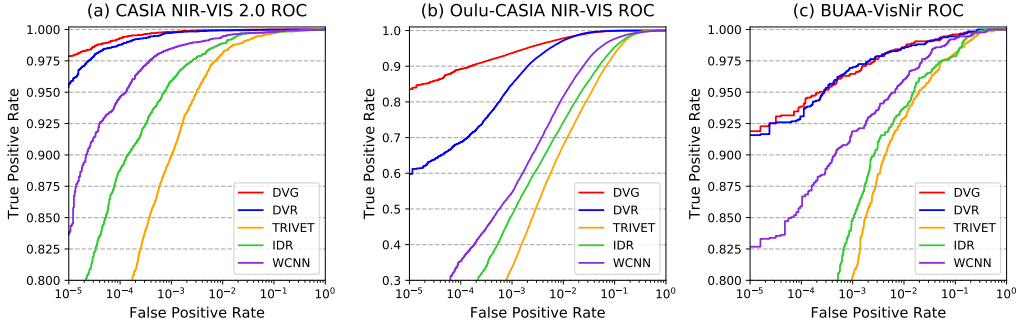

Figure 6: The ROC curves on the CASIA NIR-VIS 2.0, the Oulu-CASIA NIR-VIS and the BUAA-VisNir databases, respectively

For the most challenging CASIA NIR-VIS 2.0 database, it is obvious that DVG outperforms other state-of-the-art methods. We first employ LightCNN-9 as the backbone to perform DVG, which obtains **99.2**% on Rank-1 accuracy and **98.8**% on VR@FAR=0.1%. Further, when backbone changed to more powerful LightCNN-29, DVG obtains **99.8**% on Rank-1 accuracy and **99.8**% on VR@FAR=0.1%. Moreover, for BUAA-VisNir Face database, DVG obtains **99.3**% on Rank-1 accuracy and **97.3**% on VR@FAR=0.1%, which outperforms our baseline LightCNN-29 and other state-of-the-art methods.

To further analyze the effectiveness of the proposed DVG for low-shot heterogeneous face recognition, we evaluate DVG on the Oulu-CASIA NIR-VIS and the IIIT-D Viewed Sketch Face databases. As mentioned in section 4.1, there are fewer identities or images in these two databases. Table 2 presents the performance of DVG on these two challenging low-shot HFR databases. For the Oulu-CASIA NIR-VIS database, we observe that DVG with LightCNN-29 significantly boosts the performance from $84.9$% [35] to **92.9**% on VR@FAR=0.1%. Besides, for the IIIT-D Viewed Sketch Face database, DVG also obtains **96.99**% on Rank-1 accuracy and **97.86**% on VR@FAR=1%, which significantly outperform our baseline lightCNN-29 and state-of-the-art methods including RCN and MC-CNN by a large margin.

Fig. 6 presents the ROC curves, including TRIVET, IDR, W-CNN, DVR, and the proposed DVG. To better demonstrate the results, we only perform ROC curves of DVG trained on LightCNN-29. It is obvious that DVG outperforms other state-of-the-art methods, especially on the low shot heterogeneous databases such as the Oulu-CASIA NIR-VIS database.

Expect for the above commonly used NIR-VIS and Sketch-Photo, we further explore other potential applications, including the face recognition under different resolutions on the NJU-ID database [17] and different poses on the Multi-PIE database [8]. The NJU-ID database consists of 256 identities with one ID card image ($102 \times 126$ resolution) and one camera image ($640 \times 480$ resolution) per identity. Considering the few number of images in the NJU-ID database, we use our collected ID-Photo database (1000 identities) as the training set and the NJU-ID database as the testing set. The Multi-PIE database contains 337 subjects with different poses. We use profiles ($\pm 75^o$, $\pm 90^o$)

and frontal faces as different modalities. 200 persons are used as the training set and the rest 137 persons are the testing set (Setting 2 of [13]). On the NJU-ID database, we improve Rank-1 by $5.5\%$ (DVG $96.8\%$ - Baseline $91.3\%$) and VR@FAR=$1\%$ by $6.2\%$ (DVG $96.7\%$ - Baseline $90.5\%$) over the baseline LightCNN-29. On the Multi-PIE database, the Rank-1 of $\pm 90^o$ and $\pm 75^o$ is increased by $18.5\%$ (DVG $83.9\%$ - Baseline $65.4\%$) and $4.3\%$ (DVG $97.3\%$ - Baseline $93.0\%$), respectively. We will continue to explore more applications in our future work.

## 5 Conclusion

This paper has developed a novel dual variational generation framework that generates large-scale new paired heterogeneous images with abundant intra-class diversity from noise, providing a new insight into the problems of HFR. A dual variational autoencoder is first proposed to learn a joint distribution of paired heterogeneous images. Then, both the distribution alignment in the latent space and the pairwise distance constraint in the image space are utilized to ensure the identity consistency of the generated image pairs. Finally, DVG generates diverse paired heterogeneous images with the same identity from noise to boost HFR network. Extensive qualitative and quantitative experimental results on four databases have shown the superiority of our method.

## Acknowledgments

This work is funded by the National Natural Science Foundation of China (Grants No. 61622310) and Beijing Natural Science Foundation (Grants No. JQ18017).

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
