[Reviews · NeurIPS 2019]

Reviewer 1



UPDATE: ======== After reading other reviewers’ comments and the rebuttal, I decided to raise my score by one point from 6 to 7. I am satisfied with the effort the authors made to address my two major concerns and I recommend to accept this submission in agreement with the other reviewers. Overview/Contribution: ======== The paper proposes a dual variational autoencoder to generate synthetic training data to combat the limited data in heterogeneous face recognition. The synthetic data tries to preserve identity via identity preserving generation both in the image and embedding spaces while providing sufficient variation for the training data of the downstream recognition task. The authors claim an 18% improvement in TPR while FPR @10e-5. Strengths: ======== - Most facial recognition tasks involve certain assumptions that constrain the task into homogeneous set of inputs. Heterogeneous face recognition (HFR) is an important task for many practical applications that is attracting attention recently. Learning heterogeneous face recognition with limited dataset by generating synthetic data is interesting. - The choice of paired unconditional generation instead of image-to-image translation seem to promote inter and intra-identity diversity while preserving the identity. - The semi-supervised HFR formulation with an added L2 loss component for the generated unlabeled pair could allow sufficient training of the network while implicitly learning the identity in the case of limited training data such as this HFR. - Capturing the visual quality and identity preservation using FID and MD in addition to the recognition metric and the ablation of omitting loss components in the generation in Table I is interesting. - Evaluating the method using multiple HFR datasets helps in drawing useful conclusion. This included near IR (NIR) and visual (VIS) pair and sketch and VIS pairs. Weaknesses: =========== - Although there are multiple datasets, the variation in terms of modality is limited to either NIR-VIS and Sketch-VIS. Other heterogeneous datasets with wide variations in resolutions, cameras and environmental conditions would have made the conclusions more stronger. As the task is HFR, more modalities would have been useful. - In the formulations, there were ‘trade-off’ parameters both in the generation and recognition models yet their effect on the overall recognition is not fully explored. Ablation experiments exploring the effect of each of those terms would have helped to pinpoint from where the significant improvement was coming from. Overall, the paper reads well and the task is relevant to NeurIPS audience. However, more datasets and more ablations could have helped especially when the major contribution of the paper is paired unconditional generation improves HFR. So, I suggest the authors add another dataset that is different from the pairing explored here such as pose and resolution pairing etc.

Reviewer 2



This paper presents a new unconditional Dual Variational Generation (DVG) framework that generates large-scale paired heterogeneous images with the same identity from noise. DVG promotes the inter-class diversity and makes the generated images can be used as augmented data to optimize recognition models by a pairwise distance constraint, aiming at reducing the domain discrepancy. Extensive experiments on four HFR databases show that the proposed method can significantly improve SOTA results. - Pros: The organization, writing and presentation are clear and easy to follow. The formulas are sufficient and correct. The idea is novel, the contributions are solid and the experimental results are impressive. - Cons: 1) PIM [Zhao et al., CVPR 2018] also achieves one-to-many face generation via noise term injection, please add corresponding discussion. Some related works should be mentioned, e.g., [2], [3]. [1] Jian Zhao, Yu Cheng, Yan Xu, Lin Xiong, Jianshu Li, Fang Zhao, Karlekar Jayashree, Sugiri Pranata, Shengmei Shen, Junliang Xing, Shuicheng Yan, Jiashi Feng. Towards Pose Invariant Face Recognition in the Wild. CVPR, 2018. [2] Luan Tran, Xi Yin, Xiaoming Liu. Disentangled Representation Learning GAN for Pose-Invariant Face Recognition. CVPR, 2017. [3] Jian Zhao, Lin Xiong, Yu Cheng, Yi Cheng, Jianshu Li, Li Zhou, Yan Xu, Karlekar Jayashree, Sugiri Pranata, Shengmei Shen, Junliang Xing, Shuicheng Yan, Jiashi Feng. 3D-Aided Deep Pose-Invariant Face Recognition. IJCAI, 2018. 2) Eqn. (10) has 3 hyperparameters (\lambda_1 to \lambda_3), and Eqn. (13) has 1 hyperparameter (\alpha_1). It is unclear how to assign appropriate values to these hyperparameters. A sensitivity analysis is recommended to make this work more complete. 3) Evaluation on complexity (training & inference) is recommended, which is important for real applications. - Additional minor comments: 1) The generated paired heterogeneous face databases are highly recommended to be released to push the research frontiers heterogeneous face recognition. 2) The word "multivariate" in Line 129 Page 4 could be revised to "multi-variate" to be consistent with the same word in Line 122 Page 4. The title of Sec. 4.3 & Sec. 4.4 could be re-considered since both sections are reporting experimental results.

Reviewer 3



The manuscript is clearly presented. The experimental results on four databases show the proposed method could get improve recognition significantly. The contributions of this manuscript are significant for HFR research field.

[Author Response · NeurIPS 2019]

Thanks all the reviewers for acknowledging our contributions and their valuable comments.

**To Reviewer #1 Q1**: More modalities? **R1**: Thanks. We add experiments on the NJU-ID (different resolutions) [1] and
the Multi-PIE (different poses) [2] datasets. Details of these datasets are introduced as follows.

• The NJU-ID dataset consists of 256 identities with one ID card image ($102 \times 126$ resolution) and one camer-
a image ($640 \times 480$ resolution) per identity. Considering the few number of images in the NJU-ID dataset, we
use our collected ID-Photo dataset (1000 identities) as the training set and the NJU-ID dataset as the testing set.
• The Multi-PIE dataset contains 337 persons with different poses. We use profiles ($\pm 75^o$, $\pm 90^o$) and frontal faces as
different modalities. 200 persons are used as the training set and the rest 137 persons are the testing set.

The examples of dual generation are shown in Fig. 1. For the recognition performance, on the NJU-ID dataset, we
improve Rank-1 by $5.5\%$ (DVG $96.8\%$ - Baseline $91.3\%$) and VR@FAR=$1\%$ by $6.2\%$ (DVG $96.7\%$ - Baseline $90.5\%$)
over the baseline LightCNN-29. On the Multi-PIE dataset, the Rank-1 of $\pm 90^o$ and $\pm 75^o$ is increased by $18.5\%$ (DVG
$83.9\%$ - Baseline $65.4\%$) and $4.3\%$ (DVG $97.3\%$ - Baseline $93.0\%$), respectively. All experiments demonstrate the
effectiveness of our method in other modalities.

**Q2**: More ablations? **R2**: For the generation model, the ablations of $\mathcal{L}_{\text{dist}}$, $\mathcal{L}_{\text{ip}}$ and $\mathcal{L}_{\text{div}}$ have
been reported in Table-1. We add the ablation of $\mathcal{L}_{\text{adv}}$ in Eq. (10). That is, on the CASIA
NIR-VIS 2.0 dataset, the Rank-1 decreases $0.5\%$ if $\mathcal{L}_{\text{adv}}$ is not used. For the recognition

Figure 1: The examples of dual generation on the ID-Photo and the Multi-PIE datasets.

model, the effect of $\mathcal{L}_{\text{pair}}$ in Eq. (13) can be found in Table-2 ('+DVG' means using $\mathcal{L}_{\text{pair}}$).
All ablations reveal that each component of our method is useful. Especially for $\mathcal{L}_{\text{ip}}$, $\mathcal{L}_{\text{dist}}$ and
$\mathcal{L}_{\text{pair}}$, the Rank-1 decreases $5.5\%$, $4.9\%$ and $2.1\%$ respectively on the ablations. Moreover, our method is not sensitive
to the trade-off parameters in a large range. Please see Reviewer-2' R2 for details.

**To Reviewer #2 Q1**: The relationship between DVG and PIM? Some related works? **R1**: Thanks. The noise in PIM is
to help recover invisible details. The generated 'many' faces are required to be consistent with one ground truth. Hence,
PIM is still a conditional image-to-image translation method. As mentioned in the introduction, it faces diversity and
uniqueness limitations. Differently, our method belongs to unconditional generation. That is, we generate diverse new
paired faces from noise, which alleviates the above two limitations. We will cite these related works in our paper.

**Q2**: How to assign hyper-parameters? A sensitivity analysis? **R2**: The hyper-parameters are
set by balancing the magnitude of each loss function. Fig. 2 presents the sensitivity studies of
$\lambda_1$, $\lambda_2$ and $\lambda_3$ in Eq. (10). For $\alpha_1$ in Eq. (13), when $\alpha_1$ is set to 0.0025, 0.005, 0.01, 0.02 and
0.04, the Rank-1 is $98.9\%$, $99.1\%$, $99.2\%$, $99.2\%$ and $98.8\%$, respectively. We can observe
that our method is not sensitive to these hyper-parameters in a large range. For example, the
Rank-1 only decreases $0.3\%$ when $\lambda_1$ changes from $0.1$ to $0.4$.

Figure 2: The sensitivity studies of trade-off parameters on the CASIA NIR-VIS 2.0 dataset. The backbone is LightCNN-9.

**Q3**: Complexity? **R3**: Thanks. Our method is computationally efficient. For instance, when
using one Titan XP, training the generation model on the CASIA NIR-VIS 2.0 dataset spends
3 hours. Meanwhile, in the inference stage, generating a pair of heterogeneous faces only
needs 3.2 ms. Furthermore, training the HFR network spends 1 hour.

**Q4**: Releasing the generated faces and the writing suggestions. **R4**: Thanks. We will release our codes and the
generated data. The writing of our paper has been carefully revised according to your advice.

**To Reviewer #3 Q1**: How to tune these three parameters in Eq. (10)? **R1**: The trade-off parameters in Eq. (10) are
tuned by balancing the magnitude of each loss function. In addition, the sensitivity studies of the trade-off parameters
$\lambda_1$, $\lambda_2$ and $\lambda_3$ in Eq. (10) are shown in Fig. 2. We can observe that our method is not sensitive to these trade-off
parameters in a large range. For instance, when $\lambda_1$ changes from $0.1$ to $0.4$, the Rank-1 only decreases $0.3\%$.

**Q2**: It is suggested to report the time cost. **R2**: Thanks for your advice. Our proposed framework is computationally
efficient. For example, when using one Titan XP, training the generation model on the CASIA NIR-VIS 2.0 dataset only
needs 3 hours. Meanwhile, generating a pair of heterogeneous faces in the inference stage needs 3.2 ms. Moreover,
training the HFR network needs 1 hour.

**Q3**: Apply to other heterogeneous recognition problems? **R3**: We add experiments on other two datasets, including the
NJU-ID (different resolutions) [1] and the Multi-PIE (different poses) [2] datasets. The results show that our method
can be effectively applied to more modalities. Please see Reviewer-1' R1 for experimental details. Due to the limited
time, we will explore other heterogeneous recognition tasks in the future work.

**References**: [1] Huo et al. Heterogeneous face recognition by margin-based cross-modality metric learning. IEEE
Transactions on Cybernetics 2018. [2] Gross et al. Multi-PIE. Image and Vision Computing 2010.


[Meta-Review · NeurIPS 2019]

The paper proposes a dual variational autoencoder, used to generate new synthetic training data for heterogeneous face recognition, by preserving generation both in the image and embedding spaces and providing variation for the training data of the downstream recognition task. The authors claim a big improvement in performance. Reviewers initially were convinced on the goodness of the paper and then after the rebuttal one of the reviewer increased its rate. Thus the consensuswas reached and also the area chair agreed with the acceptance rate.